# Regional Multifractal Variability of the Overall Seismic Activity in Pakistan from 1820 to 2020 via the Application of MDFA on Earthquake Catalogs

**Aftab Alam** [1], **Dimitrios Nikolopoulos** [2,*], **Demetrios Cantzos** [2], **Muhammad Tahir** [3], **Tahir Iqbal** [1], **Ermioni Petraki** [2], **Panayiotis Yannakopoulos** [4] **and Muhammad Rafique** [5]

[1] Centre for Earthquake Studies, National Centre for Physics, Shahdra Valley Road, P.O. Box 2141, Islamabad 44000, Pakistan; aftab.alam@ncp.edu.pk (A.A.); talat.iqbal@ncp.edu.pk (T.I.)
[2] Department of Industrial Design and Production Engineering, University of West Attica, Petrou Ralli & Thivon 250, Aigaleo, 12244 Athens, Greece; cantzos@uniwa.gr (D.C.); ermionipetraki@gmail.com (E.P.)
[3] Department of Management Sciences, Comsats University, Islamabad 45550, Pakistan; m_tahir@comsats.edu.pk
[4] Department of Informatics and Computer Engineering, University of West Attica, Agiou Spyridonos, 12243 Aigaleo, Greece; pyannakopoulos@yahoo.co.uk
[5] Department of Physics, King Abdullah Campus, University of Azad Jammu and Kashmir Muzaffarabad, Azad Kashmir 13100, Pakistan; mrafique@ajku.edu.pk
* Correspondence: dniko@uniwa.gr; Tel.: +30-210-5381338

**Abstract:** The overall seismicity of Pakistan from 1820 to 2020 is analysed in terms of its multifractal behaviour. Seismic events of magnitude $M_L = 3.0$ and above are spatially clustered into four distinct groups, each one corresponding to a different region of high seismic activity. The Multifractal Detrended Fluctuation Analysis (MFDA) method applied on each cluster reveals pronounced inter-cluster heterogeneity in terms of the resulting generalised Hurst exponent and fractality spectrum, possibly due to the particular tectonic characteristics of the regions under investigation. Additional results on the variability of the Gutenberg–Richter $b$-value across the defined clusters further corroborate the uniqueness of the seismic profile of each region.

**Keywords:** earthquakes; catalog data; MFDFA; hurst exponent





## 1. Introduction

Earthquakes are extremely catastrophic natural events that can severely impact both people and property. Urban populations worldwide suffer the catastrophic consequences from the tremendous energy that is released by strong earthquakes, especially if the earthquake's epicentre is nearby. Disastrous earthquakes are unavoidable as a result of natural events, but they are very hard to forecast accurately, i.e., when and where they will occur. Therefore, finding credible seismic precursors is one of science's greatest challenges [1–8]. The topic of earthquake forecasting is still subject to debate, according to Conti et al. [9].

Earthquake occurrences have long been considered to be governed by the complex dynamics driving self-organised critical (SOC) systems [10,11]. Spatiotemporal correlations of earthquake events have been attributed to the fractal nature of the seismic sequence evolution in the space and time domain [12,13]. Arguably, the majority of research involves the study of correlations in the time domain [14–19] with a focus on cluster discovery of seismic events in time [16–18] and on inter-event time distribution modelling [19]. Exclusively spatial correlations have also been investigated [20–24], albeit to a lesser extent, in which case seismic clusters in the space domain are sought [20–22] and the inter-distance distribution of seismic events is modelled [23,24]. Joint, spatiotemporal correlation is a major research topic in earthquake research and it is usually attributed to possible fractal behaviour of seismic activity [25–27]. Some studies as in Corral et al. [16], approach the

spatial aspect of the spatiotemporal analysis by incrementally sweeping the longitude and latitude of the spatial area under investigation, regardless of the region-specific tectonic features. Bak et al. [21] and Christensen et al. [28], selectively study regions wherein significant tectonic activity has been observed.

As opposed to spatial and temporal correlations, earthquake magnitude correlations have not been studied as extensively. Some related work can be found in Lennartz et al. [19] where the earthquake magnitudes of Northern and Southern California were analysed in terms of their fractal characteristics. Similarly, Aggarwal et al. [29] and Kayal et al. [30] have investigated the magnitude sequence of seismic events in Western India, with the former analysing the fractality of the region as a whole and the latter deriving local fractal properties on small area patches comprising the whole region. Detrended Fluctuation Analysis (DFA) [31] has been extensively applied in the magnitude time series with notable examples being the work of Lennartz et al. [19] and Varotsos et al. [32] focusing on the seismic catalogs of California and the work of Varotsos et al. [15] wherein the seismic catalog of Japan is studied. MDFA [33], a generalisation of DFA, has been applied on earthquake magnitude series by Aggarwal et al. [29] in Western India and by Flores-Márquez et al. [34] in Southern Mexico.

In the present work, an extensive earthquake magnitude catalog for Pakistan during the period 1820–2020 is studied under the multifractal framework employing the MDFA method. The motivation of this work is the paper of Flores-Márquez et al. [34], in which the multifractality of a wide area is investigated by determining non-overlapping seismic zones that diachronically exhibit high seismic activity. The present work focuses on seismic clusters that are formed a-posteriory by analyzing seismic data from various catalogs for Pakistan. The Gutenberg-Richter *b*-value [35], quantifying the magnitude-frequency relation of earthquakes, is employed to differentiate the seismic profile of each cluster. To the knowledge of the authors, this is the first time that Pakistani earthquake catalog data of such a long period is utilised with the application of multifractal methods.

The following sections present, at first, the division of the studied area, vis-a-vis the seismicity distribution. Then, the method of Multifractal Detrended Fluctuation Analysis (MFDFA) is described followed by the description of the derivation of catalog data. Thereafter, the clustering of data is presented along with the related results and discussion.

## 2. Materials and Methods

### 2.1. Area of Study and Seismicity of the Period

Pakistan is situated at the junction of the three tectonic plates namely the Indian, Eurasian and Arabian plates (Figure 1). Due to the plate interactions and intra-plate movements, the geology of the country is very complex with several active faults (Figure 1). However, it can be divided into two major parts; (a) the plain region and (b) the mountainous region. The plain region covers the eastern part of the country that is subdivided into (a1) the Indus Plain and (a2) the Thar Desert. The mountainous regions include (b1) the Makran subduction zone in the southwest part of the country, (b2) the Chaman transform plate boundary in the central part of the country and (b3) the Hindukush-Himalaya-Pamir continental collision zone in the northern part [36]. Regions (a1) and (a2) are tectonically stable, hence, seismically quieter. On the contrary, regions (b1), (b2) and (b3) are seismically very active.

Since the period of study is extended (from 1820 to 2020), it was considered crucial to collect as many events as possible from the available seismic catalogs for Pakistan. The events from the seismic catalogs of the following agencies were utilised: (i) the United States Geological Survey (USGS); (ii) the Global Centroid Moment Tensor (GCMT); (iii) the International Seismological Centre (ISC) and (iv) the Centre for Earthquake Studies (CES) of Pakistan.

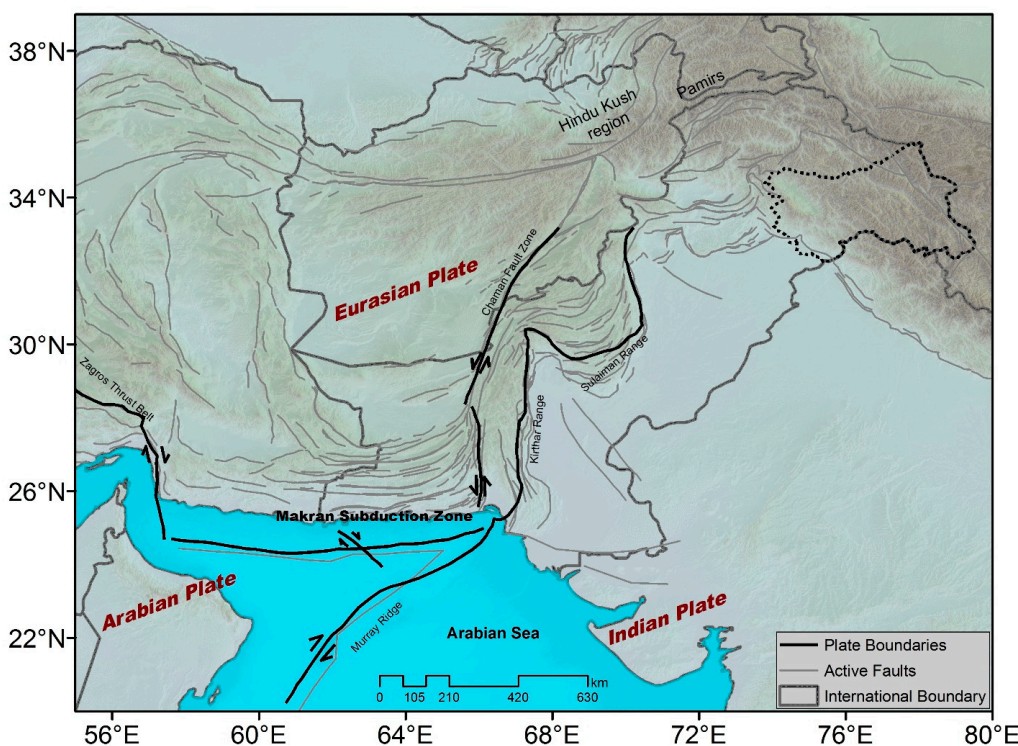

**Figure 1.** Regional and local tectonic setting of area of study.

After collecting all the data from the above sources and for all the regions (a1–b3), one unified catalog was created by merging all the data after identifying and removing duplicate events. While removing duplicate events, the seismic records of ISC that are ranked with quality A and B were given the highest priority. The CES seismic network is more detailed in the central part of the country and for this reason the events of the catalogs for this region were ranked with priority as, first priority those of CES, second priority those of USGS, third priority the records of ISC that are ranked with quality C and finally, fourth priority the records of ISC that are ranked with quality D. For the events of regions (a1), (a2), (b1) and (b3) the catalog data of ISC were ranked as first priority and of quality A or B. Second priority was assigned to the records of USGS, third priority to the records of ISC which, in addition, were ranked with quality C, fourth priority to the records of ISC which, furthermore, were ranked with quality D and, finally, fifth priority was assigned to the records of CES.

A significant issue that was addressed is that the different agencies employ different magnitude types in their records, specifically the $M_b$ (body wave magnitude), the $M_s$ (surface wave magnitude), the $M_L$ (local magnitude) and the $M_w$ (moment magnitude). This differentiation would create a significant problem in the final full seismic catalog, since records could be with different magnitude types. To overcome this problem, the moment magnitude $M_w$ was selected as the unified magnitude and all the other magnitude types were converted to $M_w$. The conversion relationships of Scordilis et al. [37], Indris [38], Ambraseys and Bommer [39], Ambraseys and Bilham [40], Ristau et al. [41] and the one developed inside CES for the corresponding CES data, were used to convert the body wave ($M_b$) and surface wave ($M_s$) magnitudes to moment magnitudes ($M_w$) (conversion $M_b M_s \rightarrow M_w$). The available moment magnitude data were not converted ($M_w \rightarrow M_w$). The different magnitude scale data were converted to moment magnitudes using the relations of Hutton and Boore [42] Mushtaq et al. [43], Mushtaq et al. [44] and Tahir et al. [45].

Finally, in order to filter out erroneous entries and weak seismic events of the generated full catalog, a limit was set to $M_w = 3.0$, i.e., seismic events with magnitudes $M_w < 3.0$ were discarded. Hence the final filtered full catalog, contains the occurred earthquakes between

1820 and 2020 with magnitudes $M_w \geq 3.0$. Hereafter, the referenced earthquakes refer to this filtered final catalog.

### 2.2. Multifractal Detrended Fluctuation Analysis (MFDFA)

The essential feature found in both monofractal and multifractal signals is that they show statistical invariance when their scale is changed. Because of this, a segment of a monofractal or multifractal signal is statistically similar to a segment of the same signal that has been produced by magnification or shrinkage at various scales. Both monofractals and multifractals may evolve in the time and/or spatial domain while exhibiting irregular fluctuations and diverging long-range correlations. The key difference between the two is that monofractals are sufficiently characterised by a single power law while multifractals require a collection of power laws with different exponents. The technique that is most frequently used to find multifractals is MFDFA.

MFDFA has a wide range of uses as in one-dimensional, discrete or continuous time series [46], mathematics [47], economics and two-dimensional data, such as maps and images [46]. Several studies [48–51], have effectively employed MFDFA.

Application of MFDFA

MFDFA identifies the scaling properties of the $q$-th order moment of a time series. MFDFA is implemented as follows [47]:

1.  The mean value of a time series $z_i$ of length $N$ ($i = 1, 2, 3 \ldots N$) is calculated as:

$$z_{avg} = \frac{1}{N} \sum_{k=1}^{N} z_{\kappa} \tag{1}$$

2.  If the time series' incremental changes around the average value follow of a random walk, the integrated profile $y(i)$ is obtained as

$$y(i) = \sum_{k=1}^{i} [z_{\kappa} - z_{avg}] \tag{2}$$

    where $i = 1, 2, 3 \ldots N$. Note that by integrating the time series the measurement noise is reduced.

3.  The time series is split into $Ns$ discrete non-overlapping bins, where $Ns$ is the integer part of $\left(\frac{N}{s}\right)$ and $s$ is the time span. Since $\left(\frac{N}{s}\right)$ is not by definition an integer, and therefore $N$ is not always an integer multiple of $s$, a small part of the time series is not taken into account, hence it is not processed. To overcome this, the same process is implemented however inversely starting from the end of the series to its beginning. In this way the non-processed segments are compensated and a better estimation is achieved.

4.  In every bin, the series's data is fitted to a polynomial and the variance $v$ is calculated in the forward ($v = 1, 2, \ldots, Ns$) and backward ($v = Ns + 1, \ldots$) directions in order to find the local trend of each of the two $Ns$ bins. Then, the square of the fluctuations is calculated as

$$F^2(s, v) = \frac{1}{s} \sum_{i=1}^{s} \{y[(v-1)s + i] - y_v(i)\}^2 \tag{3}$$

    where $y_v$ is the local polynomial fit of the integrated profile $y(i)$ at value $v$. Likewise in every bin's $v$ to the backward direction, the square of the fluctuations is:

$$F^2(s, v) = \frac{1}{s} \sum_{i=1}^{s} \{y[N - (v - N_s)s + i] - y_v(i)\}^2 \tag{4}$$

5.  After detrending the series, the $q - th$ order fluctuation function is calculated as the average of all the squares of the fluctuations in both the forward and backward directions as

$$F_q(s) = \left\{ \frac{1}{2N_s} \sum_{v=1}^{2N_s} \left[ F^2(s, v) \right]^{\frac{q}{2}} \right\}^{\frac{1}{q}} \tag{5}$$

where the exponent $\frac{1}{q}$ is a variable when $q \neq 0$ and $q$ is real. Fluctuation $F_q(s)$ in Equation (5) is defined only for spans $s \geq m + 2$.

When $q = 0$ Equation (5) yields to the logarithmic averaging according to Equation (6):

$$F_0(s) = exp\left\{ \frac{1}{4N_s} \sum_{v=1}^{2N_s} ln[F^2(s, v)] \right\} \approx s^{h(0)} \tag{6}$$

When $q = 2$ Equation (5) coincides with the typical monofractal DFA procedure.

6.  From the above equations, the generalised fluctuation functions are calculated for various $q$ values and time spans $s$. If the time series $z_i$ has long-range power-law correlations, $F_q(s)$ exhibits, for long values of scales $s$, a power law with $h(q)$ according to Equation (7):

$$F_q(s) \sim s^{h(q)} \tag{7}$$

where the exponent $h(q)$ is known as Generalised Hurst Exponent. In order to calculate $h(q)$, the $F_q(s)$ vs. $s$ is plotted in a log-log scale for as a function of $q$. When $q = 2$, the classical Hurst Exponent, $h(q = 2)$, is derived from Equation (7) and the corresponding log-log plot is the usual DFA diagram [46,47]. If $h(q)$ is independent of $q$, the time series is monofractal. If $h(q)$ is a function of $q$ the series is multifractal, because tiny and large variations scales act differently. The stronger the dependence of $F_q(s)$ with $h(q)$ the stronger the multi fractal features that the series exhibit. Note that, for negative $q$, $h(q)$ describes the scaling behaviour of segments with small fluctuations (small deviations from the corresponding fit) whereas for positive $q$, $h(q)$ describes the scaling behaviour of segments with large fluctuations (large deviations from the corresponding fit).

7.  The generalised Hurst Exponent $h(q)$ is associated with classical scaling exponent $\tau(q)$ according to Equation (8):

$$\tau(q) = q(h(q)) - 1 \tag{8}$$

Monofractal time series with long range dependencies is characterised by a linear relation between $\tau(q)$ and $q$, namely there is only a single exponent, the Hurst exponent. Multifractal time series have non-linear relation between $\tau(q)$ and $q$ and consequently, there are multiple Hurst exponents.

8.  The multifractal behaviour of time series can be delineated through the multifractal spectrum $f(\alpha)$ versus $\alpha$, where $\alpha = (d\tau)/(dq)$ is the Legendre transform of $\tau(q)$, $f(\alpha) = q\alpha - \tau(q)$. Note that $\alpha$, known the Holder exponent, estimates the singularity strength, while $f(\alpha)$, specifies the fractal dimension of the subset series, that is characterised by $\alpha$.

The association of $\alpha$, $f(\alpha)$ and $h(q)$ are summarised in Equations (9) and (10):

$$\alpha = h(q) + qh'(q) \tag{9}$$

$$f(\alpha) = q[\alpha - h(q)] - 1 \tag{10}$$

The plot of $f(\alpha)$ versus $\alpha$ (singularity spectrum) is the most commonly used approach to outline the multifractal behaviour of time series.

9.  Each singularity spectrum is fitted by a quadratic function at the point of its maximum at $\alpha_0$. This quantifies the intensity of the multifractal behaviour of the singularity spectrum because it measures the range of the multifractal exponents that are present in each plot. It is for this reason that is referred many times as the degree of fractality. Extrapolating the fitted quadratic curve to zero, the spectrum's width $W$ is calculated. The richer the multifractality in the dataset, the wider the width is [48,49]. By definition $W$ is given by Equation (11) [50]:

$$W = a_{max} - a_{min} \tag{11}$$

### 2.3. Frequency Magnitude Distribution

The existence of fractal dimension in nature over fault zone drive magnitude distribution as a power law For finite time and specific region earthquakes, frequency magnitude distribution decays as a power law called Gutenberg-Richter law [35]. If $N(M)$ is a number of earthquakes of magnitude $M$ above $M_c$, then frequency magnitude distribution (FMD) can be expressed as:

$$N(M) = 10^{a - b \cdot M} \tag{12}$$

whereas, parameter $a$ is the intercept of FMD and represents activity rate. The computation of this parameter is required in hazard assessment studies [52]. The parameter $b$ is the slope of FMD and depicts the relative abundance of large to small events. This scaling parameter is extensively used to elaborate different aspects of earthquakes [53–58]. Although for a long time window, its value is close to unity $b \simeq 1$, significant variations have been observed for shorter time window, in the range of $b = 0.5$ to $b = 2.5$ [59,60]. Typically, FMD is affected by several factors and finding the mechanisms which are responsible for these fluctuations is still under debate. However, many studies attribute these variations to changes in stress conditions [61–63]. The relationships of Utsu et al. [64] and Utsu et al. [65] were used to estimate $b$-values by using average magnitude ($\overline{M}$) and magnitude interval/bin ($\Delta M = 0.1$) as

$$b = \frac{1}{ln(10) \cdot \left[ \overline{M} - \left( M_c - \frac{\Delta M}{2} \right) \right]} \tag{13}$$

where $M_c$ is the completeness magnitude.

If $N$ is a the total number of events then standard error in $b$-value can be calculated by using Shi and Bolt [61] as

$$\Delta b = \frac{b}{\sqrt{N}} \tag{14}$$

## 3. Results and Discussion

The filtered seismic catalog for Pakistan that was created from the four different source catalogs of USGS, GCMT, ISC and CES (Section 2.1) contains 2394 seismic events from 1820 to 2020. When this data were grouped in space regardless of the time of their occurrence, the event vectors were interestingly spatially grouped into four non-overlapping clusters, each of which, significantly, corresponded to a high seismicity region or, equivalently, a high seismic event density. The spatially grouped seismic events of the full filtered catalog and the associated clusters are presented in Figure 2.

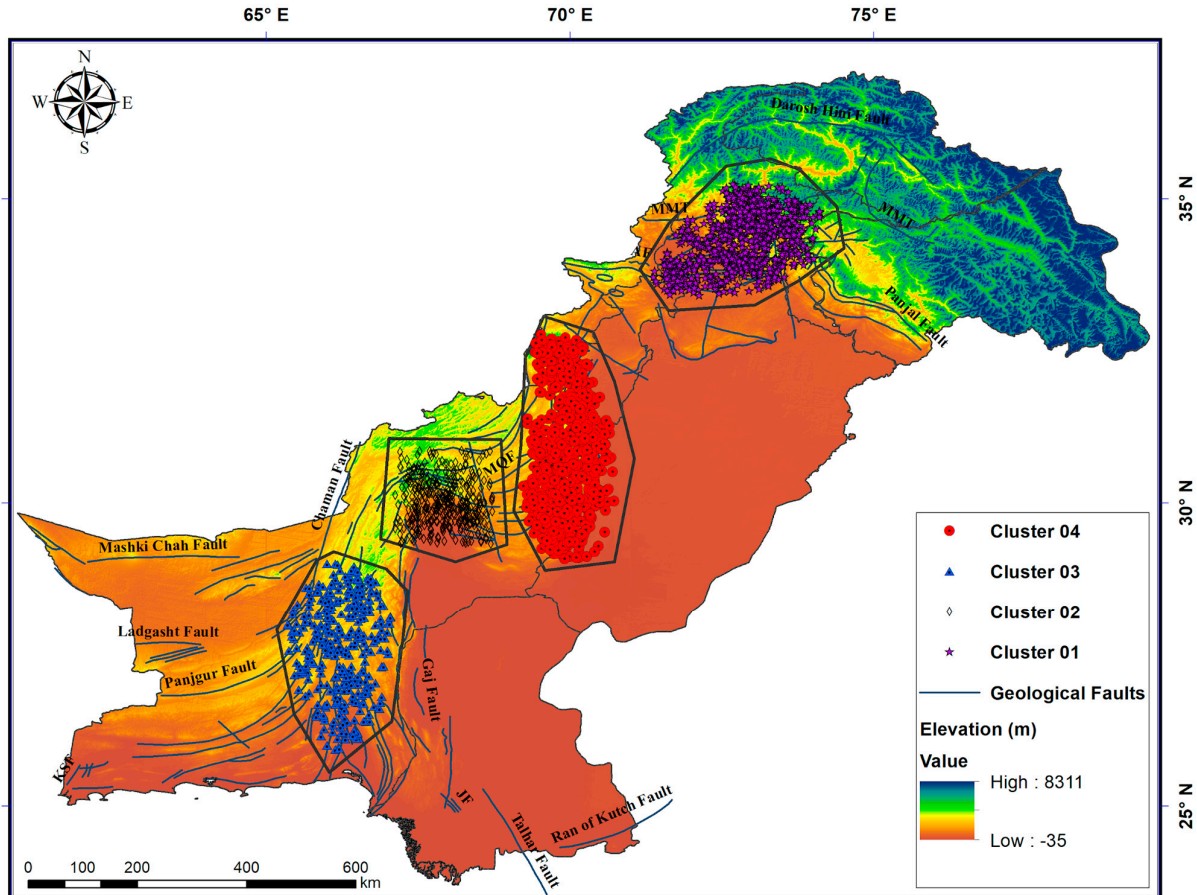

**Figure 2.** The locations of the four seismic clusters under investigation in central and southwestern Pakistan. Elevation data and the major faults are also depicted.

This is not the first study that identifies clustering of spatially grouped seismic events. Seismic clusters are reported in a very recent study [62] around the Moho transition zone in the subduction zone of the Pacific plates and the Tethys collision zones. Seismic clusters are also found in two recent studies by Yamagishi et al. [63,66]. The growth of clusters is reported recently by Fischer and Hainzl [67], while the seismic clusters in Okinawa are reported by Arai [68]. Seismic clusters have been reported in past papers as well [17,18,22,34,55,68–71].

Regarding specific cluster data, cluster 01 is in the northwest part of the country and owes its seismicity to the continental collision between the Indian and Eurasian tectonic plates. This results in the formation of the great mountain massifs of Pakistan which include Hindukush, Karakoram, Pamir and Himalayas (see also Figure 1). As far as the tectonics of the north Pakistan concerns, this can be subdivided into several areas [72]. These areas are from south to north: The Salt Range, the Potwar and Kohat plateaus, the Hill Ranges, the intermontane basins, the southern Kohistan Ranges, the Nanga-Parbat-Haramosh regions, the Main Mantle Thrust, and the Kohistan island arc, which is separated from Asian rocks of the Pamirs to the north by the Main Karakorum Thrust. Seismically north Pakistan is very active and the faults within fold-and-thrust belt have been frequently producing moderate to large seismic events that include the $M_w = 7.6$ Kashmir earthquake of 2005 that resulted in more than 75,000 lives loss in addition to a colossal economic loss [43–45,73–78].

On the other hand, clusters 02, 03 and 04 cover different areas of Central Pakistan where the regional tectonic movements are mainly governed by the left lateral transform boundary between the Indian and the Eurasian plates. This transform boundary is named as Chaman Fault System and traverses two collision zones, namely, the Makran Subduction Zone in the south and the Hindukush-Pamir continental collision zone in the north. To

the east of the main transform fault, a foreland fold and thrust belt runs parallel to it for, almost, the whole of the length. The fold and thrust belt is dominated by tight folds and overthrusts making the highlands of Sulaiman and Kirthar mountain ranges [79,80]. Although earthquakes occur all along the transform boundary zone, more intensive seismic activity is observed along specific structures created due to sharp bends in the fold and thrust belt [40,81,82]. For example significant earthquakes, such as the $M_w = 7.3$. Mach earthquake of 1931 and the $M_w = 7.6$, Quetta earthquake of 1935 can be associated with Quetta Syntaxis [40], while the $M_w = 7.7$ Awaran earthquake of 2013 occurred near the junction of multiple segments belonging to this fault system [83].

The number of great earthquakes in each cluster as well as the total number of seismic events varies between the four clusters of Figure 2 (Table 1). Cluster 01 contains one earthquake with magnitude $M_w > 7.0$. This earthquake is the 2005 Kashmir earthquake. The Kashmir earthquake is considered very important because it contains the largest aftershock sequence of Pakistan ever recorded. Cluster 01 contains also the highest number of total events (912) in comparison to the other clusters due to the longer time period that it includes. Cluster 02 has one great earthquake of magnitude $M_w = 6.9$ that occurred in the Sulaiman ranges. Cluster 02 contains 730 seismic events. Cluster 03 contains a great earthquake of magnitude $M_w = 7.2$ occurred in Quetta in year 1935. It also contains the great earthquake occurred also in Quetta, however, in year 1997 with magnitude $M_w = 7.1$. This cluster consists of 387 seismic events. Finally, cluster 04 includes the great Awaran earthquake of $M_w = 7.8$ of the year 2013 and an event of magnitude $M_w = 7.6$ occurred in 1935. The total number of events recorded in cluster number 04 are 365. Some of these details are also shown in Table 1.

**Table 1.** Results of completeness analysis for the entire catalog for Pakistan showing catalog time completeness for different magnitudes.

| Magnitude | Completeness Period |
|---|---|
| $M_w > 3.5$ | 2005–2020 |
| $M_w > 4.0$ | 2005–2020 |
| $M_w > 4.5$ | 1972–2020 |
| $M_w > 5.0$ | 1962–2020 |
| $M_w > 5.5$ | 1956–2020 |
| $M_w > 6.0$ | 1922–2021 |

The cumulative method is implemented here for the calculation of the completeness periods. By using such a method, a simple graph is usually plotted between the cumulative number of earthquakes versus time for a specific magnitude range (e.g., $M_w \geq 4.0$ or $M_w \geq 6.0$). The catalog is considered complete (for this particular magnitude range) with respect to time when there is roughly a straight line of the data used. In this case, the completeness period will be the number of years from the start of this straight-slope part until the last year of the catalog. Completeness periods and threshold magnitudes were estimated for the entire catalog. The cumulative distribution of earthquakes above different magnitude levels (3.5, 4.0, 4.5, 5.0, 5.5, and 6.0) with respect to the time. Completeness periods for different magnitude intervals have been tabulated in Table 1. Results show that the present catalog is complete for different magnitudes; 3.5, 4.0, 4.5, 5.5 and 6.0 are respectively 2005, 2005, 1972, 1962, 1956 and 1922 with seismicity rates of 554.33, 249, 67.71, 13.93, 4.55 and 1.66 events/year, respectively. Earthquakes with $M_w \geq 3.5$ and $M_w \geq 4.0$ are complete only for approximately the last 15 years, whereas earthquakes with $M_w \geq 5.5$ and $M_w \geq 6.0$ are complete for the last 64 and 98 years, respectively.

As can be observed from Table 1, the average $b$-values of Equation (13) and $M_c$ of the law of Equation (13) do not differ significantly ($p < 0.01$) between the various clusters. However when only one standard deviation is considered the tendency of a higher Gutenberg Richter $b$-value can be observed in cluster 02. Parameter $M_c$ differs only for cluster 01, potentially, due to the greater number of events that this cluster includes.

The cumulative number of seismic events within the various clusters are presented in Figure 2. Great discrepancies can be observed. These could be attributed to the different geological settings of each cluster as well as the proximity of clusters 01 and 02 to active faults. Other reasons for the inter-cluster discrepancies are the different total number of events of each cluster as well as the varying magnitude completeness ($M_c$) of the Gutenberg Richter law across the clusters. Indeed this completeness varies from north to south because the stations coverage of CES is better in northern part as compared to that of southern Pakistan.

The $b$ and $M_c$ values of the Gutenberg Richter law are presented in Figure 3. Differences are observed among the various clusters. This could be attributed to various physical parameters. At first, Ahmad et al. [84] and Imoto et al. [85] have observed that the stress-buildup and the strain hardening-softening result in changes of the $b$-values of the Gutenberg-Richter law. According to Scholz [57], the $b$-value is inversely proportional to the stress, thus lower $b$-values may depict higher differential stress in a region before the occurrence of the mainshock. In addition, low $b$-values have been associated with the existence of asperities over a fault plane, while the nucleation process of the earthquakes normally ruptures these locked parts or asperities [86]. Schorlemmer et al. [87] proposed a technique for determining the stationarity of $b$-values of the Gutenberg-Richter law, which was further used for the probabilistic earthquake forecasting across the San Andreas Fault. On the basis of micro-seismicity size distribution prior to the Parkfield event of $M_w = 6.0$ a low $b$-value in the area of highly stressed patches was observed by Schorlemmer and Wiemer [53]. All these concepts can explain the observed discrepancies in Figure 3.

Figure 4 presents the fluctuation functions $Fq(s)$ for as a function of the scale $s$ per cluster data according to Equation (7) for $q$ values ranging between $-10$ and $10$ with step 5. As with Figures 3 and 5, the input data for Figure 4 are the seismic events of the full filtered catalog per cluster. The fluctuation functions exhibit linear increasing trends with scale $s$. The reader should recall here that since $Fq(s)$ is a power law of scale $s$, the slope of the linear associations of Figure 4 is the generalised Hurst exponent $h(q)$. It is observed that the higher $q$-values are shifted up and this tendency is seen in the results of every cluster, that is for every seismic cluster. This changing with $q$ implies that the $M_w$ magnitude series are multifractals and therefore follow non-linear patterns. This in turn suggests that the pure statistical analysis of seismic sequence data, as the one presented in Figures 3 and 5, and Table 1, is of limited character and can not outline all trends in similar datasets.

It is very interesting and has to be emphasised that the MFDFA curves of $Fq(s)$ versus $s$ for clusters 03 and 04 in Figure 4 are very similar. The reader should recall in relation from Figure 2, that these clusters are near active geological faults with similar geological settings and this is reflected in the MFDFA results from the seismic data of these clusters. The MFDFA curve of cluster 01 is comparable with the curves of clusters 03 and 04 for $Log_{10}(S)$ values greater than 1.22. A reason for the similarities and discrepancies of the MFDFA curves of cluster 01 is that this cluster is located on the Main Boundary Thrust, which, on the one hand, is an important seismic geology, but, on the other hand, different from the geology of clusters 03 and 04. Another possible reason for the above discrepancies might also be that several earthquakes of cluster 01 occurred in higher elevations. The MFDFA curve of cluster 02 is different. These tendencies are also observed in Figure 6 in the curves of the generalised Hurst exponent $h(q)$ versus $q$ and in the curves of $\tau(q)$ versus $q$ in Figure 7. To retrieve both curve variations with MFDFA, the relations of $Fq(s)$ versus $s$ were computationally generated for $q$ values of step 1. The generalised Hurst exponent $h(q)$ was calculated as the slope of the batch of these least square fits and $\tau(q)$ was calculated from the $h(q)$ values according to Equation (8). The $h(q)$ versus $q$ curves (Figure 6), as well as, those of $\tau(q)$ versus $q$ (Figure 7) are similar for clusters 03 and 04. Cluster 01 has similar curves shapes with the ones of clusters 03 and 04, both for the $h(q)$ versus $q$ variations (Figure 6) and for the $\tau(q)$ versus $q$ ones, but with a different value range. On the contrary the corresponding curves for cluster 02 are different. Therefore it can be supported from the MFDFA data of all subfigures of Figures 4, 6 and 7 that the proximity of clusters 03 and

04 to active faults and the similarities but differentiations of geological settings of cluster 01, may explain the MFA trends. These tendencies could not have been identified on a pure statical basis as already mentioned above.

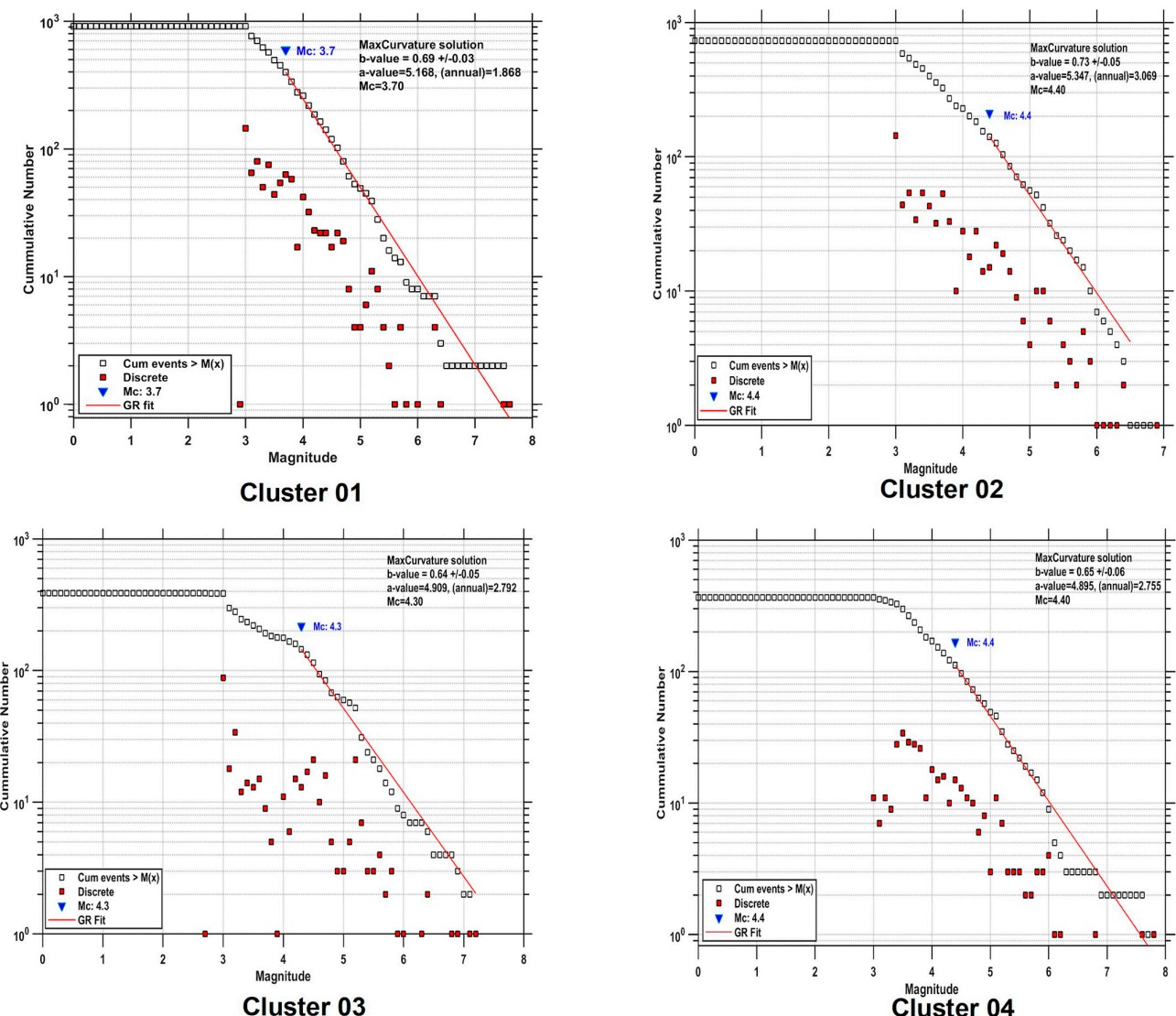

**Figure 3.** Plot of the Gutenberg Richter's law *b*-value (slope) versus the Magnitude ($M_w$). The magnitude completeness $M_c$ per cluster is also given. The data is derived from the full filtered seismic catalog per cluster. The subfigures were generated with the ZMAP software version 7 [88].

Fluctuation functions $Fq(s)$ as a function of the scale *s* for earthquake data are reported by Flores-Márquez et al. [34] for seismic five zones in Mexico, by Chamoli and Yadav [72] for seismic series in NW Himalaya, by Telesca et al. [48] for seismic series in Italy, by Telesca et al. [49] for seismic interspike series in Italy, for other earthquake related series [12,89–93] and for other types of series [50,94]. As with the $Fq(s)$ versus *s* curves of Figure 4, Flores-Márquez et al. [34] report increasing tendencies of $Fq(s)$ versus *s* both for the Guerrero earthquake magnitudes time series, as well as in each one of the five seismic zones. Comparable increasing trends of $Fq(s)$ versus *s* are also reported by Telesca et al. [48,49], Telesca and Lapenna [91], in other papers for seismic data [12,92,93] and in urban pollution MFDFA data series [94]. In similar plots of $h(q)$ versus *q* as those of Figure 6, Flores-Márquez et al. [34] reports comparable $h(q)$ versus *q* with overlap. Such overlap exists also in the curves of Figure 6 If these curves are linearly approximated, the slopes of $h(q)$ versus *q* for clusters 03 and 04 are approximately 0.02 while the, roughly estimated, linear slope of the plots in Flores-Márquez et al. [34] is

approximately 0.175. Telesca and Lapenna [90] also report comparable curves shapes with overlap. Telesca et al. [49] report $h(q)$ versus $q$ as those of cluster 02 of Figure 6 for seismic interspike series in Italy. The corresponding $h(q)$ versus $q$ curves reported by Telesca et al. [93] are quite different.

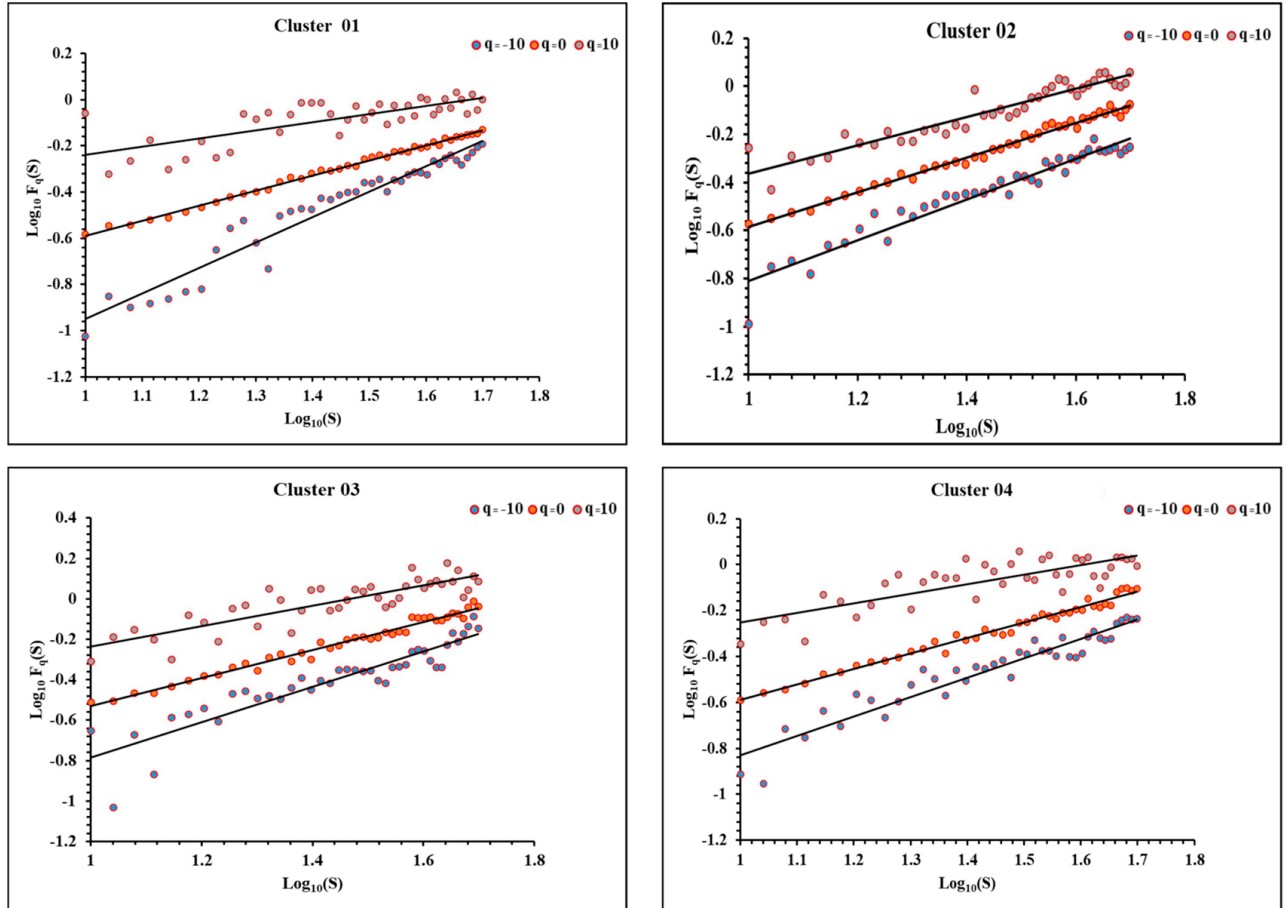

**Figure 4.** Fluctuations functions $Fq(s)$ for $q = -10$, 0 and 10 as a function of the scale $s$ per cluster data according to Equation (7) and corresponding linear trends.

Figure 8 presents the multifractality spectra of all clusters according to Equations (9) and (10). It should be emphasised that the origins of multifractality in the time series of the sub-figures of Figure 8, may result from (a) the existence of a broad probability function (b) the varied contributions of the small and big fluctuations to the overall long-range correlations (c) or from a combination of (a) and (b) [49,50,94]. The value of $\alpha_0$ that corresponds to the maximum of the $\alpha$, $f(\alpha)$ curve, measures the process' regular or irregular behaviour. The larger values of $\alpha_0$ suggest that the process will be comparatively more regular [72]. In addition, the width $W$ of Equation (12) provides information on the symmetry of the multifractal spectra. Spectra that are right or left skewed are associated to the weighting of high or low fractal exponents [72]. The range of $\alpha_0$ and $W$ are crucial parameters for interpretation.

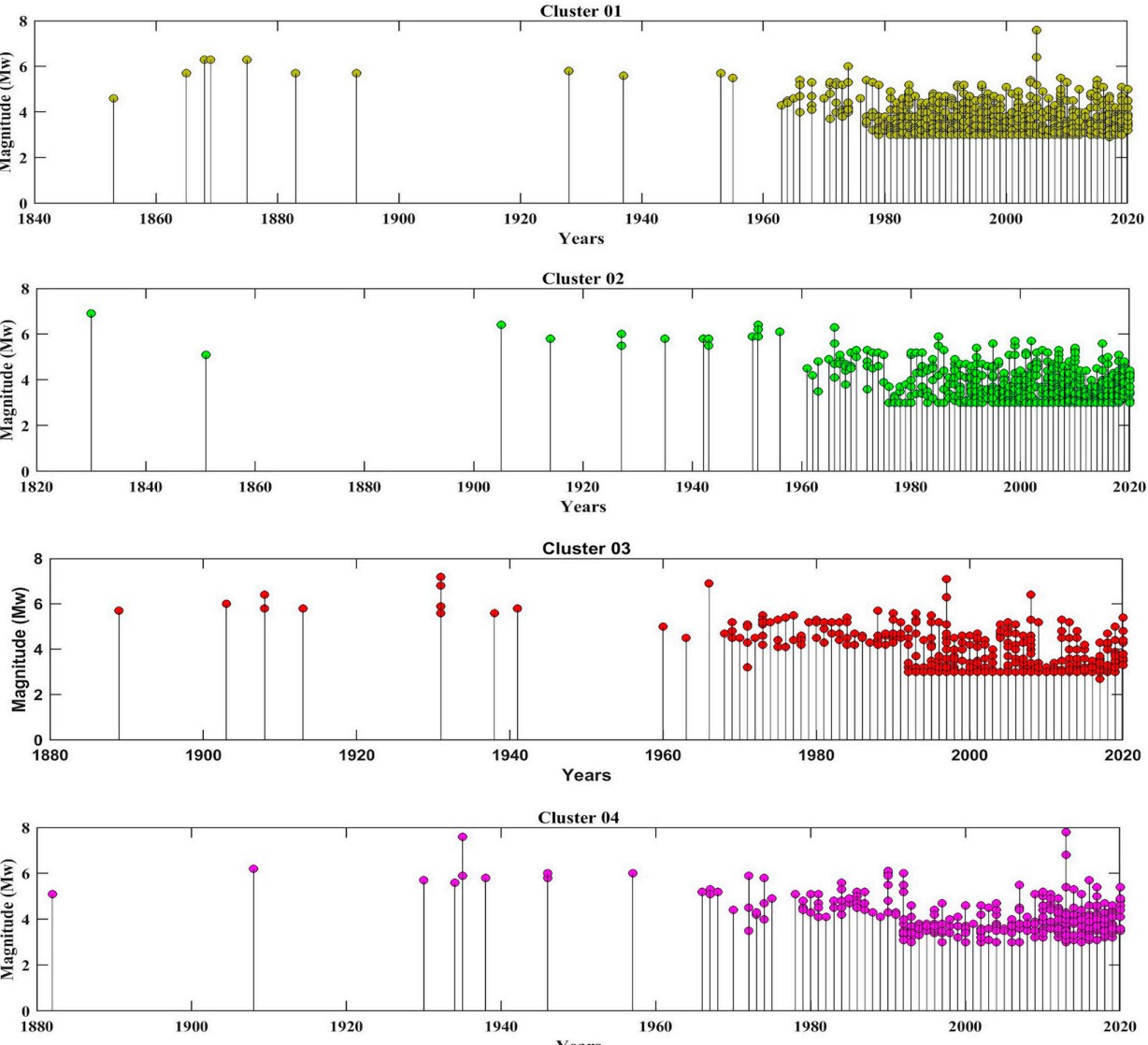

**Figure 5.** The number of events with magnitude $M_w$ of the full filtered catalog between 1820 and 2020 per cluster. Earthquake temporal distribution from different clusters from 1820–2020, that comprised historical, pre-instrumental and instrumental earthquakes before 1906, 1906–1975 and after 1976 respectively.

As also identified and emphasised in Figures 4, 6 and 7 the multifractal spectra of clusters 03 and 04 are similar and different from the ones of clusters 01 and 02. Significantly, the multifractal spectrum of cluster 02 is symmetrical. According to the above, this implies equal weight of the low and the high fractal exponents to the seismic data of cluster 02. This could be explained by the rather fault-clear geological underground and the fact that all occurred seismic events are in, similarly, shallow depth events. The spectrum of cluster 01 is also rather symmetrical. The fact, however, that the occurred earthquakes are spatially distributed in various elevation levels (i.e., at various depths), could explain the slight non-symmetry, in comparison to the one of cluster 02. The most important finding is however that the spectra of clusters 03 and 04 are right skewed. Therefore the high fractal exponents play a more important role in the seismic data of these clusters. This could be attributed to the proximity of these clusters to active geological faults of Pakistan, as also have been observed from the other outcomes of this paper.

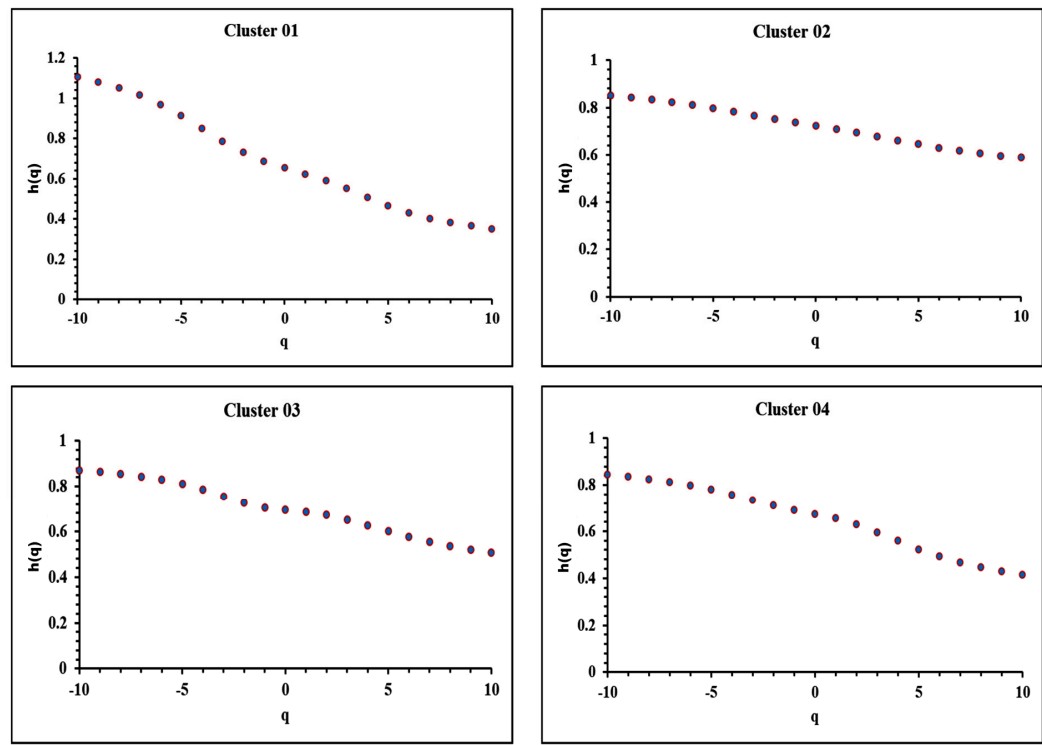

**Figure 6.** The generalised Hurst exponents of each cluster data versus $q$.

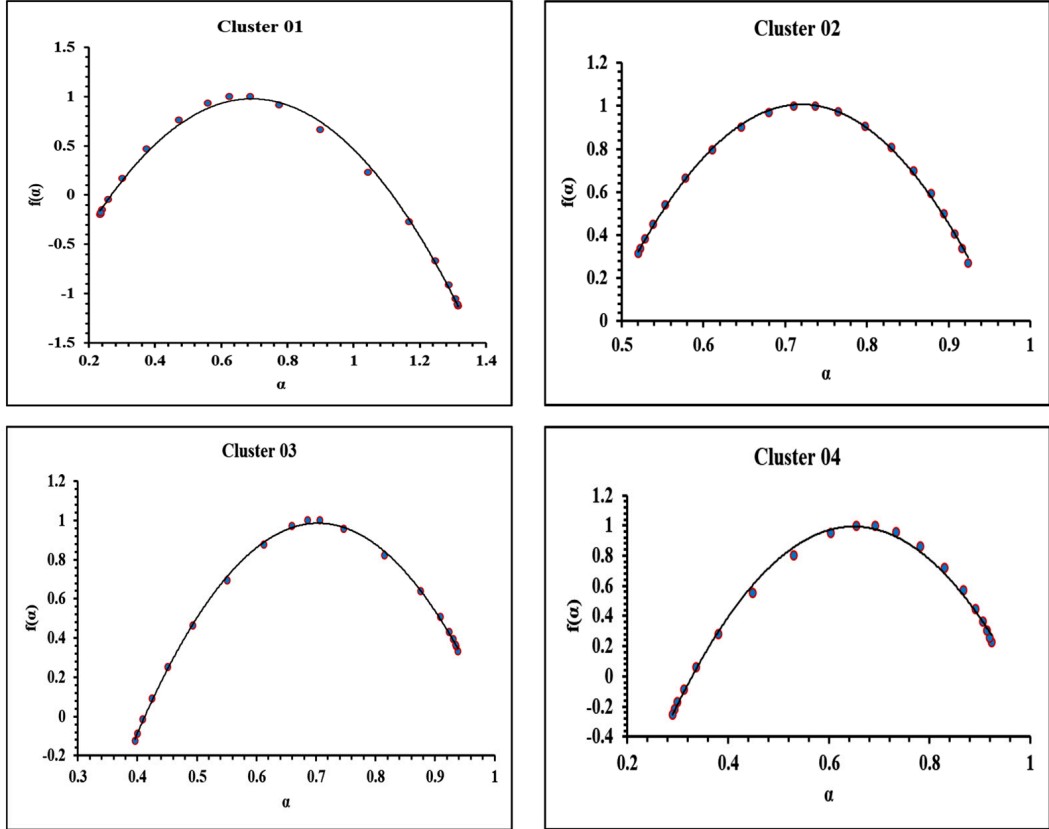

**Figure 7.** Curves of parameter $\tau(q)$ versus $q$ calculated from Equation (8) from the data of each cluster.

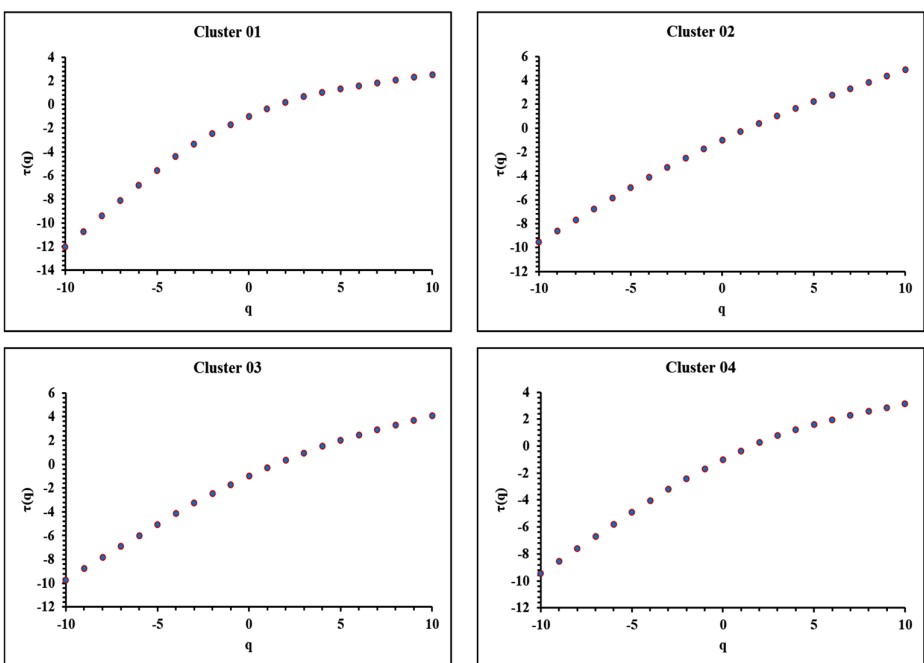

**Figure 8.** Multifractal spectra for each cluster.

The multifractal spectra of clusters 03 and 04 are very similar to those reported by Flores-Márquez et al. [34] for the four of the five zones of Mexico. They are also similar to the ones reported by Barman et al. [92]. On the other hand the $f(\alpha)$ plots reported by Telesca et al. [48] and by Telesca and Lapenna [91] for seismic sequences in Italy are very different. The multifractal plots reported by Chamoli and Yadav [72] for seismic series in NW Himalaya are also different.

## 4. Conclusions

The present paper reports statistical results via the Gutenberg Richter law and multifractal analysis outcomes via MFDFA for four seismic clusters identified from earthquake events that occurred in Pakistan from 1820 to 2020. In order to build the earthquake event database, the earthquakes from four different catalogs were accessed and reorganised to provide a full filtered seismic catalog for Pakistan containing the moment magnitudes ($M_w$) of 2394 earthquakes. The four clusters are located in different geological settings. Clusters 03 and 04 have similar geology, cluster 01 is located on the Main Boundary Thrust of Pakistan. The Gutenberg Richter $b$ and $M_c$ values differentiate between the clusters due to the different geology and data completeness. The application of the Gutenberg Richter law within each cluster, showed that the majority of seismic events of each cluster, and especially the very destructive earthquakes, are described by the Gutenberg Richter law and this verified the validity of the full filtered catalog of Pakistan. Further analysis is reported through MFDFA in each cluster data. The fluctuation functions $Fq(s)$ for as a function of the scale $s$ per cluster showed significant multifractal patterns present in the earthquake data of each cluster. The generalised Hurst exponents $h(q)$ versus parameter $q$ and in the curves of $\tau(q)$ versus $q$ showed similar multifractal trends in the data of clusters 03 and 03 and comparable mutiftractal patterns in the data of cluster 01. These tendencies were found also in the multifractal spectra of clusters 03 and 04 versus the data of cluster 01. All results related to the Gutenberg Richter law are within the international ranges of the literature. The multifractal results are comparable to the published data which are however limited. The geological settings of each cluster are discussed in association with the presented outcomes. This is the first such paper for Pakistan.

**Author Contributions:** Conceptualization, A.A.; methodology, A.A., D.N., D.C., M.R., M.T., T.I. and E.P.; software, A.A.; formal analysis, D.N., D.C., A.A., M.R., M.T., T.I., E.P. and P.Y.; investigation, A.A., D.N., D.C. and P.Y.; resources, A.A., M.R., M.T. and T.I.; writing—original draft preparation, D.N., D.C., A.A., M.R., M.T. and T.I.; writing—review and editing, D.N., D.C., A.A., M.R., E.P. and P.Y.; visualization, A.A.; supervision, A.A. and M.R.; project administration, D.N. All authors have read and agreed to the published version of the manuscript.

**Funding:** This research received no external funding.

**Data Availability Statement:** Data sets are available by the international agencies mentioned in the references in text.

**Conflicts of Interest:** The authors declare no conflict of interest.

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
