# Peer review of "Regional Multifractal Variability of the Overall Seismic Activity in Pakistan from 1820 to 2020 via the Application of MDFA on Earthquake Catalogs"

_fractalfract, doi:10.3390/fractalfract7120857_

Round 1

Reviewer 1 Report

Comments and Suggestions for Authors

The paper “Regional Multifractal Variability of the Overall Seismic Activity in Pakistan from 24 B.C to 2020 via the Application of MDFA on Earthquake Catalogs” by Alam et al. presents the multifractal analysis of earthquake timeseries in Pakistan based on the MDFA method, for four spatial seismic clusters along the active fault zones where seismicity is focused. As it is mentioned in the text, this is the first such study for Pakistan that experiences high seismicity and large magnitude and destructive earthquakes. The paper presents some interesting results and the methodology is sound. However, there are some points in the data analysis and structure that need improvement and probably revision.

The main one is the dataset. The authors attempt to homogenize the seismic catalog by converting the different magnitude scales to moment magnitude based on various relations that exist in the literature. The range of the catalog, however, spanning back to 24 B.C. to include the great Taxila earthquake can create artifacts. The authors estimate the magnitude of completeness for cluster 1 as Mc=3.7. The latter means that the catalog is complete for earthquakes with magnitude ≥3.7 since 24 B.C., which is definitely not the case. Figure 5, that shows the cumulative number of events with time, is informative enough to point out that the catalogs for the 4 clusters are complete for the reported, by the authors, Mc only during the last decades. So, I would suggest a revision in the manuscript regarding the dataset that is used.

Another issue with the manuscript is Section 3, which is too long. I would suggest breaking it into subsections, where the seismic data, the frequency-magnitude distribution analysis, the application of MDFA to the data and the Discussion are distinctly presented.

Some minor issues concern:     

1) In Fig.3, the cluster number in each subplot should be provided.

2) In Page 9 it is mentioned “The cumulative number of seismic events within the various clusters are presented in Fig.2.” I think this is done in Fig.5. 

3) The analysis in Fig.4 seems to be executed with ZMap. If this is the case, provide the appropriate reference. Furthermore, the caption of Fig.4 says, “Plot of the Gutenberg Richter’s law b-value (slope) versus the Magnitude (Mw).” This is the solid fitted line, while the plots show the cumulative number of events with magnitude.

4) The same for Fig.5. The caption says, “Cumulative number of earthquakes of the Gutenberg Richter law with Magnitudes (Mw)”, while it shows the cumulative number of events with time for each cluster.

5) It is not clear in the text to which data is the MDFA applied to. Is it the magnitude timeseries?

6) In Page 12 the authors say, “Another possible reason for the above discrepancies might also be that several earthquakes of cluster 01 occurred in higher elevations.” Do they refer to the topography of the area? The hypocentres’ depths are more relevant to the physical process of earthquakes rather than the elevation.  

Comments on the Quality of English Language

English language in the text can be improved is some cases.

Author Response

We would like to thank the reviewer for reading and commenting our manuscript. We appreciate the valuable time spent. 

To assist the reconsideration of our paper we provide two discrete files additional to the whole revision:

  1. MultifractalPakistan_Alam_et_al_v23_cm.pdf: This pdf shows the corrections as red text. The deletions are shown as strikethrough text.
  2. MultifractalPakistan_Alam_et_al_v23.pdf This pdf is the final clean paper

Below is our response to the comments. This document is part of the cover letter and also uploaded separately for convenience.

We hope that we responded adequately

Response to comments Reviewer 1

Comment 1:

The paper “Regional Multifractal Variability of the Overall Seismic Activity in Pakistan from 24 B.C to 2020 via the Application of MDFA on Earthquake Catalogs” by Alam et al. presents the multifractal analysis of earthquake timeseries in Pakistan based on the MDFA method, for four spatial seismic clusters along the active fault zones where seismicity is focused. As it is mentioned in the text, this is the first such study for Pakistan that experiences high seismicity and large magnitude and destructive earthquakes. The paper presents some interesting results and the methodology is sound. 

Reply 1:

We would like to thank the reviewer for this substantial outline of our work

Comment 2: 

However, there are some points in the data analysis and structure that need improvement and probably revision. 

Reply 2:

We have tried to address all issues indicated by the reviewer. We hope that we have implemented our revision adequately.

Comment 3:

The main one is the data set. The authors attempted to homogenize the seismic catalog by converting different magnitude scales to moment magnitude based on various relations that exist in literature. The range of catalog however, spanning back to 24 BC to include the great Taxila earthquake creates artifact.  

Reply 3:

This is a valuable comment and it was an overestimation by our side. Indeed a large range of earthquake data may create artefacts. We believe that the selection and processing of catalog data, is an important step for quality assessment, especially in terms of magnitude and time. A complete and consistent catalog of earthquakes can provide good data for studying the distribution of earthquake in certain region as a function of space, time and magnitude. 

After this comment, we re-evaluated our catalog data. Consequently, in the revised manuscript, we have used the dataset that is complete in both time and magnitude, while the incomplete part of the dataset was ignored. In the initial manuscript, the great Taxila earthquake of 24 B.C was included in catalog processing (part of the comment referring to homogenisation and completeness), but this earthquake was not included in fractal analysis and b-value estimation.

Indeed, the historical data may violate the time completeness for different magnitude bins. We apologise that we had not considered that in our initial manuscript and, as a result, the completeness period of the catalog was wrong in our initial manuscript. 

In the revised paper, a new paragraph is added (sixth of the results section) and Table 1 was rewritten. In this revised paper, we consider only the catalog data that is complete for specific time and magnitude bins. 

Accordingly, we have also changed the title of the manuscript.

We hope that we responded adequately.

Comment 4:

The authors estimate the magnitude of completeness for cluster 1 as Mc=3.7. The latter means that the catalog is complete for earthquake with magnitude ≥ 3.7 since 24 B.C., which is definitely not the case.

Reply 4:

This is a very valuable comment and we thank the reviewer for that. As indicated in our Reply 3, a new paragraph is added (sixth of the results section) and Table 1 was rewritten. Therefore the Mc-value of 3.7 for cluster 1, was estimated for the complete catalog (without the the Taxila earthquake of 24 B.C., please see initial manuscript). As mentioned by the reviewer, latter earthquake was of 7.5 magnitude, but this magnitude is beyond the cut-off limit of magnitude > 6.0 for the completeness period of 1922-2022. 

For clarifying this, we have created the following figure (Fig.1) only for this reply. It is evident that this catalog was not complete for earthquakes before 1900 and this justifies why the Taxila Earthquake was not included.

pastedGraphic.png

Figure 1: Completeness analysis of earthquake catalog from the cumulative number of earthquakes above magnitude interval Mw > 6.0

Moreover, the CES seismic network started in 1976.  Initially six short-period stations were installed at the centre of Pakistan. Later on, in the year 2005, all stations were replaced with broadband seismometers. The majority of seismic stations operate in northern Punjab Pakistan, including the Kohat-Potwar Plateau and Punjab plains, with a few stations operating in the southern part of Pakistan. Thus, events primary and secondary azimuthal coverage in central Pakistan is greater than 180o. The southwestern part of the country (Baluchistan) has poor seismic coverage (two out of three seismic stations are in working condition). Thus, the magnitude of completeness (equation.pdf) varies between central and southern parts of Pakistan due to the distribution and configuration of seismic stations. Accordingly, the Mc-value is relatively lower for Cluster 1 compared to Cluster 4 due to the different number of seismic stations that were in working condition. The number of events, the event detection, and threshold magnitude are strongly dependent on the number of seismic stations. A region with a higher number of stations has a higher number of events recorded and, thus, lower threshold magnitude. Our used catalog was complete in both time and magnitude, while the incomplete part of the dataset was ignored.

We hope that we responded adequately.Comment 5:

Figure 5, that show the cumulative number of events with time is informative enough to point that the catalog for four clusters is complete for the reported, by the authors, Mc only during the last decades. So i would suggest the revision in the manuscript regarding the data set that is used.

Reply 5:

As indicated in our Reply 4 and Reply 5, we added a new paragraph which reflects our aspects on this issue. We believe that the data are adequate for the period indicated in the Title, that is after 1820 and surely not only for the last decades. We believe that in the revised manuscript the issues are clarified in the manner that reflects also our aspects and the reviewers comments.

We hope that with our previous replies, Figure 1 of this response and the text in our revised manuscript, are enough to cover this comment. 

We hope that we have responded adequately.

Comment 6:

Another issue with the manuscript is Section 3, which is too long. I would suggest breaking it into subsections, where the seismic data, the frequency-magnitude distribution analysis, the application of MFDFA to the data and discussion are distinctly presented.

Reply:

We understand the suggestion of the reviewer. This has with the way of writing. However, we have a different way. That is we separate the Results and Discussion  section in different paragraphs that are organised as much as they could be organised. We mean, that it is not possible to discretise into sub-sections, since several things are discussed in many parts, onwards and backwards. This is also reflected in the flow of the references numbers. There are references from the first parts of the beginning of the 3rd section, in the middle and the last parts. Another issue is that this, we believe, would distract the reader much, since we have sub-sections in the methods part (which can be done so). Finally, we have the opinion, that a text with sub-sections in the results and discussion gives the impression of a draft report and not a scientific paper.

We are very sorry that our writing aspects differ in this comment.

Minor Issues

Comment 7:

In Fig. 3, the cluster number in each subplot should be provided.

Reply 7: 

We provided the cluster number in each subplot in revised draft of manuscript.

Comment 8:

In Page 9 it is mentioned” The cumulative number of seismic events within the various clusters is presented in Fig. 2,”I think this is done in Fig 5.”

Reply 8:

Yes it was a mistake. We have found it. Since we deleted in the revised manuscript Figure 5, we corrected this part as well.

Comment 9:

The analysis in Fig. 4 seems to be executed with ZMap. If this is the case, provide the appropriate reference. Furthermore, the caption of Fig. 4 says, “Plot of the Gutenberg Richter’s Law b-value (slope) versus the magnitude (Mw).” This is the solid fitted line, while the plots show the cumulative number of events with magnitude.”

Reply 9

The ZMap software was used for estimating and plotting of b-value. We added reference 85 for ZMap software and renumbered the remaining references. We also added this information to the caption of Figure 4

Comment 10:

The same for Fig.5. The caption says, “Cumulative number of earthquakes of the Gutenberg Richter’s Law with Magnitude (Mw)”, while it shows the cumulative number of events with time for each cluster.

Reply 10:

We deleted Figure 5, so there is no similar issue now.

Comment 11:

It is not clear in the text to which data is the MFDFA applied to. Is it the magnitude time series?

Reply 11:

Yes MFDFA was implemented on magnitude column of dataset used for current research work. We believe it is clear.

Comment 12:

In page 12 the authors says, “Another possible reason for the above discrepancies might also be that several earthquakes of cluster 01 occurred in higher elevations. “Do they refer to the topography of the area? The hypocenters depths are more relevant to the physical process of earthquakes rather than the elevation.”

Reply 12:

We agree that the hypocenters depths are more relevant to the physical process of earthquakes rather than the elevation. Since the phrase “higher elevations” that was used in initial manuscript was inappropriate so we have changed the sentence to “shallow depth events” and implemented appropriate rephrasing to the text.

Reviewer 2 Report

Comments and Suggestions for Authors

Dear Authors,

I read with interest the paper, and appreciated the efforts and work that I can see is behind it. My first concern is the completeness of the catalogue: the number of earthquakes is small for such a long time period and from the analysis of the Gutenberg Richter curves you obtain Mc>>3. Hence I would redo all the analyses taking the data above the Mc, and it would be preferable to use the same Mc for the whole region. Then you can compare the results and eventually interpret them in terms of different seismic behaviour and characteristics of the fault. I would also suggest to comment in terms of seismic behaviour of the various tectonic zones or seismogenic fault.

But before this, looking at the diagrams magnitude versus index of the events (correct?) I see that the magnitude is higher for the first 60-70 events. Probably they correspond to the oldest part of the catalog. If so, there is probably a bias in the conversion from intensity to Mw. Hence, you should discard these events before performing analyses of the distribution of the magnitude and perform the analysis starting from a year later than what you used now. Eventually, treat separately the two intervals, but I fear the number of earthquakes in the oldest part is too low.

Then, apart from a few comments you find in the annotated manuscript, you should add a bit of discussion about the meaning of multifractality in general, and in your case in particular. What is the meaning of the results? Do they help us to understand some peculiarity of the seismicity of the various clusters? Are the clusters characterized by bursts of seismicity, long quiescences, or, on the contrary, regular seismicity or ...?

Comments on the Quality of English Language

Apart from a couple of mistakes, the language is good. I commented a couple of sentences to be rewritten, however, because they are obscure or not clear enough.

Author Response

We would like to thank the reviewer for reading and commenting our manuscript. We appreciate the valuable time spent. 

We have taken into consideration all comments raised and those given in the annotated pdf file. We appreciate the effort for the enhancement of our paper.

To assist the reconsideration of our paper we provide two discrete files additional to the whole revision:

  1. MultifractalPakistan_Alam_et_al_v23_cm.pdf: This pdf shows the corrections as red text. The deletions are shown as strikethrough text.
  2. MultifractalPakistan_Alam_et_al_v23.pdf This pdf is the final clean paper

Below is our response to the comments. This document is part of the cover letter and also uploaded separately for convenience.

We hope that we responded adequately

Response to comments Reviewer 2

Comment 1:

I read with interest the paper, and appreciated the efforts and work that I can see is behind it.

Reply 1:

We would like to thank the reviewer for this kind evaluation of our work.

Comment 2:

My first concern is the completeness of the catalogue: the number of earthquake is small for such a long time period and from the analysis of the Gutenberg Richter curves you obtain Mc>>3. Hence I will redo all the analysis taking the data above the Mc, and it would be preferable to use the same Mc for the whole region. Then you can compare the results and eventually interpret them in terms of different seismic behavior and characteristics of the fault.

Reply 2:

We appreciate and thank the reviewer for this comment. 

Indeed, as the reviewer indicated, the number of earthquakes was small when compared to the long time period that was presented in the initial manuscript. Especially for the periods for which seismic instrumentation was not available. 

After this comment, we re-evaluated our catalog data. Consequently, in the revised manuscript, we have used the dataset that is complete in both time and magnitude, while the incomplete part of the dataset was ignored. However in the initial manuscript, the great Taxila earthquake of 24 B.C was included in catalog processing (part of the comment referring to homogenisation and completeness), but this earthquake was not included in fractal analysis and b-value estimation. We concluded that the historical data may violate the time completeness for different magnitude bins. For this reason the completeness period of the catalog was wrong in our initial manuscript. Please accept our apologies for that.

In the revised paper, a new paragraph is added (sixth of the results section) and Table 1 was rewritten. In this revised paper, we consider only the catalog data that is complete for specific time and magnitude bins. 

Accordingly, we have also changed the title of the manuscript.

Moreover, the CES seismic network started in 1976.  Initially six short-period stations were installed at the centre of Pakistan. Later on, in the year 2005, all stations were replaced with broadband seismometers. The majority of seismic stations operate in northern Punjab Pakistan, including the Kohat-Potwar Plateau and Punjab plains, with a few stations operating in the southern part of Pakistan. Thus, events primary and secondary azimuthal coverage in central Pakistan is greater than 180o. The southwestern part of the country (Baluchistan) has poor seismic coverage (two out of three seismic stations are in working condition). Thus, the magnitude of completeness (Μc) varies between central and southern parts of Pakistan due to the distribution and configuration of seismic stations. Accordingly, the Mc-value is relatively lower for Cluster 1 compared to Cluster 4 due to the different number of seismic stations that were in working condition. The number of events, the event detection, and threshold magnitude are strongly dependent on the number of seismic stations. A region with a higher number of stations has a higher number of events recorded and, thus, lower threshold magnitude. Our used catalog was complete in both time and magnitude, while the incomplete part of the dataset was ignored.

We have considered the reviewer's idea of using the same Mc-value for all clusters but in that case, the number of events will be further reduced from the complete part (magnitude completeness and obeying power law) of the catalog. As we use the complete catalog, obeying a power law in terms of magnitude, increasing the Mc-value does not change our results regarding the b-value, as all used data follow a straight line (Figure 1). On the same time, as mentioned above, less data is considered with no specific gain.  

pastedGraphic.png

Figure 1 : Changing Mc-value within the power law, depicts the same b-value. This is created only for this response and not included in the paper.

We hope that we have responded adequately.

Comment 3:

I would also suggest to comment in terms of seismic behavior of the various tectonic zones or seismogenic fault.

Reply 3:

We think that the discussion of the initial manuscript and the revised manuscript, covers as much as the existing data allow. We believe that we do not have aimed experiments to differentiate the various tectonic maps. Therefore a comment on that may raise ambiguities and questions. Our study is, more ore less, a-posteriori, that is after the events have occurred. However, since this is a significant issue, we will try to address it in the future. 

Comment 4:

Looking at the diagrams magnitude versus index of the events (corrects?) I see that the magnitude is higher for the first 60-70 events. Probably they correspond to the oldest part of the catalog. If so, there is probably a bias in the conversion from intensity to Mw. Hence you should discard these events before performing analyses of the distribution of the magnitude and perform the analysis starting from a year later than what you used now. Eventually, treat separately the two intervals, but I fear the number of earthquakes in the oldest part is too low

Reply 4: 

As discussed in our Reply 2, we re-evaluated the catalog data and made the changes indicated there. We agree in all parts of this comment, that is also a reason for making all these changes. 

We hope that we have responded adequately.

Comment 5:

Then, apart from a few comments you find in the annotated manuscript, you should add a bit of discussion about the meaning of multifractality in general, and in your case in particular

Reply 5:

In the beginning of section 2.2.we have an introduction of what multifractal mean. We could write far many things. We include after this paragraph, a collection of our papers and papers from other teams that mention many things about fractals, multifractals and the related interpretations. The problem is that if we mentioned much details, the reader would be distracted from the first part of the research, that is the justifications based on the GR  law. In section 3, we include a full discussion in all paragraphs after former Fig.5 caption, where we discuss what each figure means and the literature that supports our claims. Under these aspects, we believe that we have compensated both parts of the manuscript in the following sense: a) Present GR law  data (corrected now) that support that our catalog data is complete for further study (could be just this one, a single paper); b) Use the complete GR justified data, to utilise multifractals and investigate what can be found (several things are reported). The main conclusion is that the GR is not enough, because the earth’s crust during fracture has fractal and mutifractal paths. Therefore the behaviour via GR law, needs further data for the investigation. However, as we have mentioned in our reply here, even this combination can not support unambiguous geological interpretations, because focused experiments are needed. On the other hand, it presents very useful data for all the EQ related scientific community. More ore less, we summarise our aspects here for clarity. All these are distributed inside the text, in the way that we write. Below you may find our papers and some useful papers from other teams on interpretations of fractals and mutifractals (only to outline, how many more things could have been written, or may be written)

For example we include here our papers focused in fractals and also multi fractals.

Papers from our team: 

  • 10.1007/s10967-013-2764-8,
  • 10.1039/c3ay26486f,
  • http://dx.doi.org/10.1016/j.apradiso.2012.09.005,
  • http://dx.doi.org,10.4172/2157-7617.1000244,http://dx.doi.org
  • 10.1080/19475705.2014.945496,
  • 10.4172/2157-7617.1000367,
  • 10.4172/2157-7617.1000359,
  • https://doi.org/10.1007/s10950-018-9781-6,
  • 10.4172/2157-7617.1000465,10.2343/geochemj.2.0571,
  • http://dx.doi.org/10.3390/geosciences10060235,
  • https://doi.org/10.1016/j.jastp.2021.105775,
  • https://doi.org/10.3390/geosciences13090268 

as well as in air pollution series 

  • http://dx.doi.org/10.3390/environments6030029,
  • https://doi.org/10.1007/s00703-020-00744-3,
  • https://doi.org/10.3390/e23030307,
  • https://doi.org/10.3390/environments10010009). 

We have not mentioned these and some others papers of our team, to avoid self-citations.

Papers from other teams: 

  • Earth Planets Space, 54, 1237–1246, 2002,
  • Nonlinear Processes in Geophysics (2003) 10: 1–14,
  • 10.1016/j.pce.2003.11.012,
  • 10.1103/PhysRevLett.92.065702,
  • Earth Planets Space, 57, 215–230, 2005,
  • 10.1007/s11069-006-9021-1,
  • PHYSICAL REVIEW E 71, 066123 (2005),
  • www.nat-hazards-earth-syst-sci.net/6/205/2006/,Tectonophysics 431 (2007) 273–300,
  • Tectonophysics 431 (2007) 263–271,
  • Nonlin. Processes Geophys., 15, 379–388, 2008,
  • PHYSICAL REVIEW E 77, 036101 (2008),
  • Physica A 387 (2008) 1161–1172, 
  • Nat. Hazards Earth Syst. Sci., 9, 1953–1971, 2009 ,
  • Nat. Hazards Earth Syst. Sci., 10, 275–294, 2010,
  • Physica A 389 (2010) 133–140,Chaos, Solitons and Fractals 19 (2004) 1–15, Physics and Chemistry of the Earth 29 2004, 445-451,
  • Natural Hazards and Earth System Sciences, 5, 673–677, 2005, 
  • Tectonophysics 423 (2006) 115–123,
  • http://dx.doi.org/10.1063/1.4879519

and many others.

We hope that our reply covered your kind suggestions and that we have responded adequately.

Comment 6:

What is the meaning of results? Do they help us to understand some peculiarity of the seismicity of the various clusters? Are the clusters characterized by burst of seismicity, long quiescence, or, on the contrary, regular seismicity or?

Reply 6:

We think that we have covered this issue in our Reply 5. 

Regarding the first part of the question, yes, they help us understand that the GR is not enough. The multifractals add more results and identify patters that could not be discovered through the GR law. Yes, therefore, this is a peculiarity of the clusters since the multifractal behaviour of the various  clusters was different.

Regrading the second part of the question, now. No. We can not support any of these partial implications. No bursts of seismicity, since our approach is over a long period, and also not through GR comparison of  between different chronological periods (not  also with multifractals as well). Of course the long quiescence can not be supported by the views expressed in this paper. Since the majority of the earthquakes followed the GR law it may be supported that is regular seismicity. However, if the above papers of our team, it can be seen that we support that there is no one-to-one correspondence of earthquakes and seismic patterns. Moreover each strong earthquake occurs over a great geological area. 

In order not going into long details, you may check our last paper in Geosciences (Alam, Aftab, Dimitrios Nikolopoulos, and Nanping Wang. 2023. "Fractal Patterns in Groundwater Radon Disturbances Prior to the Great 7.9 Mw Wenchuan Earthquake, China" Geosciences 13, no. 9: 268. https://doi.org/10.3390/geosciences13090268). In this paper we employ 13 different fractal techniques to address issues as those implied in this comment. Also our paper of 2020 (Nikolopoulos, Dimitrios, Ermioni Petraki, Panayiotis H. Yannakopoulos, Georgios Priniotakis, Ioannis Voyiatzis, and Demetrios Cantzos. 2020. "Long-Lasting Patterns in 3 kHz Electromagnetic Time Series after the ML = 6.6 Earthquake of 2018-10-25 near Zakynthos, Greece" Geosciences 10, no. 6: 235. https://doi.org/10.3390/geosciences10060235), addresses the post activity signs. And that only via EM radiation. A review of our team outline the issues for both EM radiation and radon (Fractal Analysis of Pre-Seismic Electromagnetic and Radon Precursors: A Systematic Approach (DOI: 10.4172/2157-7617.1000376).

We tried, in the present paper, to keep the views at a more “standard” way, that is GR law and multifractals. We believe that we compensated between these. Of course we are open to reconsider our views and expressions if more details ares needed. Therefore, if the apexes above are not satisfactory, we may proceed in expressing a lot more details on the interpretations of multifractals. 

We hope that we have responded adequately. 

Round 2

Reviewer 2 Report

Comments and Suggestions for Authors

Dear Authors,

thank you for having addressed all my comments, and corrected the manuscript accordingly, by performing additional work. Now your results are convincing. Personally, I think that comments on the seismic behaviour of the seismic zones could have been added, even if I understand your points, but it is a matter of personal interests and opinion. Your paper for me can be accepted for publishing.